

# Difficulties when using video playback to investigate social cognition in California scrub-jays (*Aphelocoma californica*)

Katharina F. Brecht[1,2], Ljerka Ostojić[1], Edward W. Legg[1] and Nicola S. Clayton[1]

[1] Department of Psychology, University of Cambridge, Cambridge, United Kingdom
[2] Institute of Neurobiology, University of Tübingen, Tübingen, Germany

## ABSTRACT

Previous research has suggested that videos can be used to experimentally manipulate social stimuli. In the present study, we used the California scrub-jays' cache protection strategies to assess whether video playback can be used to simulate conspecifics in a social context. In both the lab and the field, scrub-jays are known to exhibit a range of behaviours to protect their caches from potential pilferage by a conspecific, for example by hiding food in locations out of the observer's view or by re-caching previously made caches once the observer has left. Here, we presented scrub-jays with videos of a conspecific observer as well as two non-social conditions during a caching period and assessed whether they would cache out of the observer's "view" (Experiment 1) or would re-cache their caches once the observer was no longer present (Experiment 2). In contrast to previous studies using live observers, the scrub-jays' caching and re-caching behaviour was not influenced by whether the observer was present or absent. These findings suggest that there might be limitations in using video playback of social agents to mimic real-life situations when investigating corvid decision making.

Corresponding author
Katharina F. Brecht,
katharinabrecht@gmail.com

## INTRODUCTION

The use of video stimuli offers a high degree of flexibility in the presentation of stimuli in animal experiments as it allows for a controlled presentation of inanimate objects as well as conspecifics and social agents in general. Furthermore, the possibility to alter certain features allows the experimenter to create well-suited control stimuli (*D'Eath, 1998*; *Rieucau & Giraldeau, 2009*). This characteristic of video clips can be especially helpful when assessing socio-cognitive processes in animals. Here, an important advantage of using video over live stimuli is that video clips allow contrasting social and non-social stimuli congruently. Moreover, it is possible to present the same stimuli repeatedly or to different subjects, whereas presenting live conspecifics (or other social agents) is difficult to control in such situations. A recent example of the application of videos is a study by *Sliwa & Freiwald (2017)* assessing neuronal networks of social interactions in rhesus macaques. To unravel which brain regions were specifically dedicated for social interactions, fMRI scans were made of four macaques watching videos of conspecifics interacting together and contrasted to videos of an individual conspecific acting alone as well as with objects

moving and interacting (and additional controls). Additionally, the use of video playback allowed for the precise control of the timing of the presentation of stimuli, which ensured average activations could be calculated across the subjects (*Sliwa & Freiwald, 2017*). Hence, videos allow for a controlled investigating of social cognition.

Birds of the crow family are well-known for their socio-cognitive abilities (e.g., *Clayton & Emery, 2015*). Here too, videos could provide a useful means for manipulating and displaying social stimuli to investigate their socio-cognitive abilities in more detail, for example, in order to assess which cues elicited by a conspecific the birds are sensitive to. However, the suitability of video playback in behavioural experiments with birds in general has been contested (*D'Eath, 1998*). There are a suite of limitations regarding the presentation of video stimuli to birds, stemming from the fact that video screens are built for the human not the avian visual system (*D'Eath, 1998*; *Lea & Dittrich, 2000*). One example is the critical flicker-fusion frequency (CFF): The continuous perception of individual screens as a video differs between animals, with humans perceiving a video with 25 frames per second as continuous, while for example zebra finches have a CFF of 55.3 Hz (*Crozier & Wolf, 1941*) and therefore may perceive the same video as flickering as opposed to continual, seamless motion. While there are studies confirming that these limitations pose a problem for the suitability of video playback with birds (*D'Eath & Dawkins, 1996*; *Ryan & Lea, 1994*), there are also reports of successful use to display conspecifics **to** birds in different contexts. For example, in chickens, video playback of conspecifics produces audience effects (*Evans & Marler, 1991*) and affects feeding behaviour via social facilitation (*Keeling & Hurnik, 1993*). Male Zebra finches react appropriately to the display of a female conspecific, even to subtle changes in behaviour of the female (*Galoch & Bischof, 2007*) and use the feeding choice of a demonstrator bird presented as video playback to guide their own feeding decision (*Guillette & Healy, 2017*). Video playback of a group of conspecifics produced similar effects on foraging in nutmeg mannikins to when real conspecifics were present (*Rieucau & Giraldeau, 2009*). Additional examples include studies in starlings (*Zoratto et al., 2014*), quail (*Ophir & Galef, 2003*) and junglefowl (*McQuoid & Galef, 1993*).

In corvids in particular, a recent study reported that Eurasian jays were able to transfer learned associations about live objects to videos of these objects (*Davidson et al., 2017*). However, this study involved only inanimate objects. In the social domain, *Bird & Emery (2008)* showed that rooks prefer observing their partner over an unfamiliar conspecific, both when presented live and as video playback. Similar results have also been obtained with California scrub-jays (*Brecht, 2017*). However, it remains unclear whether the birds perceived conspecifics displayed as video playback as relevant social stimuli (i.e., as their partner).

While the rooks showed the same preference in the video and the live conditions, it is possible that they looked more at the video of the partner because they saw certain cues associated with their partner and a preference for these cues over other cues may be possible even if the birds did not interpret the bird in the video as their partner. Additionally, while birds seem to be sensitive to what is presented, it is not known to what extent they make use of this information. For example, while blue tits seem to be responsive to a conspecific presented on screen, it is less clear whether they use the social information provided by this

conspecific (*Hämäläinen et al., 2017*). Studies showing bird species changing their mating or feeding behaviour (e.g., *McQuoid & Galef, 1993*; *Rieucau & Giraldeau, 2009*) in response to a video of conspecifics could be due to the perception of certain cues that reflexively trigger such responses, similar to an artificial red stimulus that is sufficient in eliciting begging behaviour in herring gulls (*Tinbergen & Perdeck, 1950*). Recognising conspecifics as such does not imply that they are also interpreted as relevant for the current behaviour. Consequently, in order to use video playback as a means to study social cognition in the avian taxa, it is necessary to investigate whether birds, and corvids in particular, perceive conspecifics presented as video playback as relevant social stimuli that impact the subjects' behaviours and decision-making.

In order to assess this question, we used California scrub-jays' propensity to protect their caches from their conspecifics. Like other corvids, scrub-jays cache excess food for later consumption (*Vander Wall, 1990*). This cached food is at risk of being pilfered by conspecifics, whose observational spatial memory allows them to efficiently pilfer caches they have seen others make (*Watanabe & Clayton, 2007*). To protect their caches from being pilfered by a conspecific observer, scrub-jays have been found to employ a suite of different behavioural strategies (*Dally, Clayton & Emery, 2006*; *Grodzinski & Clayton, 2010*). Specifically, when observed by a conspecific, scrub-jays preferentially cache in locations that limit the observer's visual access to the cache site, such as in the shade, behind barriers, or far away from the observer (*Dally, Emery & Clayton, 2005*). If they cannot obscure the location of their caches from the observer at the time of caching (*Dally, Emery & Clayton, 2005*), scrub-jays move previously made caches to different, novel locations (re-caching) once the potential threat has left the scene (*Dally, Emery & Clayton, 2006*; *Emery & Clayton, 2001*). Similar behaviours have been reported for other corvid species, namely ravens (e.g., *Bugnyar, 2011*; *Bugnyar & Heinrich, 2005*), Clark's nutcrackers (*Clary & Kelly, 2011*) and Eurasian jays (e.g., *Legg & Clayton, 2014*; *Shaw & Clayton, 2012*). Importantly, such strategies are employed in response to social cues (e.g., *Clary & Kelly, 2011*), and seem to be specific to situations in which caching is observed by a conspecific competitor (e.g., *Dally, Emery & Clayton, 2010*; *Thom & Clayton, 2013*).

In two experiments, we presented scrub-jays with video playback of a conspecific observer while they had the opportunity to cache. We assessed two different cache-protection strategies: caching out of the observer's view (Experiment 1) and re-caching once the observer has left (Experiment 2). If scrub-jays interpreted the conspecific observer in the video as a threat to their caches—like they do with live conspecific observers—they should selectively protect their caches, i.e., preferentially cache out-of-view (Experiment 1) and re-cache (Experiment 2) when the video showed a conspecific than when the video showed an empty cage or a rope hanging from the cage ceiling (non-social conditions).

## EXPERIMENT 1

In Experiment 1, scrub-jays were presented with videos of an observer (*Observer condition*). In addition, we tested the birds in two non-social conditions: in one, the birds were presented with a video of an empty cage (*Empty Cage condition*) and in the second

condition, birds were presented with a video of a lightly swinging rope hanging from the cage ceiling (*Rope condition*). This second non-social condition was run to ascertain that a difference between the Observer and the Empty Cage condition would not be due to movement on the screen.

While they were presented with one of the three videos, scrub-jays could cache food in two different locations—one "in-view" of the video screen and one "out-of-view", behind an opaque barrier. The set-up was similar to methods previously used to assess cache-protection strategies in scrub-jays, in that scrub-jays were able to chose between a caching site in-view and a caching site out-of-view of the observer (e.g., *Dally, Emery & Clayton, 2010*). If birds are responding to video playback of the conspecific similarly to how they respond to live observers (reviewed in *Clayton, Dally & Emery, 2007*; *Grodzinski & Clayton, 2010*), they should cache more in the "out-of-view" than in the "in-view" location when presented with the video of the conspecific observer. Alternatively, if the birds showed no difference in their caching behaviour between the observer and the non-social conditions, this would suggest that—unlike in a 'live' situation—they did not perceive the conspecific in the video as a threat to their caches.

## Method
### Subjects
Five female and five male adult scrub-jays were tested between December 2014 and August 2015. Scrub-jays were housed at the Sub-department of Animal Behaviour, University of Cambridge. The work was conducted under the UK Home Office project licence PPL 80/2519. During the experimental period, subjects were housed in pairs in three to four joined cages (each measuring $1 \times 1 \times 1$ m). When not being tested birds rested in outside aviaries.

### Procedure
In a within-subject design, each bird participated in all of the three conditions and completed three separate test trials on three consecutive days. Hence, each bird saw each video once. Trial order was pseudo-randomised across birds.

Individuals could either cache in a location hidden behind the opaque side of a Perspex barrier ("out-of-view" location) or behind a transparent side of a Perspex barrier ("in-view" location). Which side of the barrier was opaque was counterbalanced across subjects but held constant across conditions.

Two hours prior to testing, individuals were separated from their cage partner, and maintenance diet was removed to ensure that birds were mildly hungry and thus motivated to cache or eat at the time of testing. Birds were tested in cage 1 (see Fig. 1A). At the start of a trial, dividers to cage 2 were removed and the screen, mounted onto the back wall, was showing one of the three videos. Birds were given a 15-minute long test phase during which they could cache. Birds were then released into their home cage and caching trays were removed for a 15-minute long break. During this break, caches were counted and placed back in the tray. Any caches made outside the tray were removed. Finally, birds were again allowed into cage 1 to retrieve any caches made previously, a procedure that has been successfully used to prevent extinction of caching in previous caching studies (reviewed

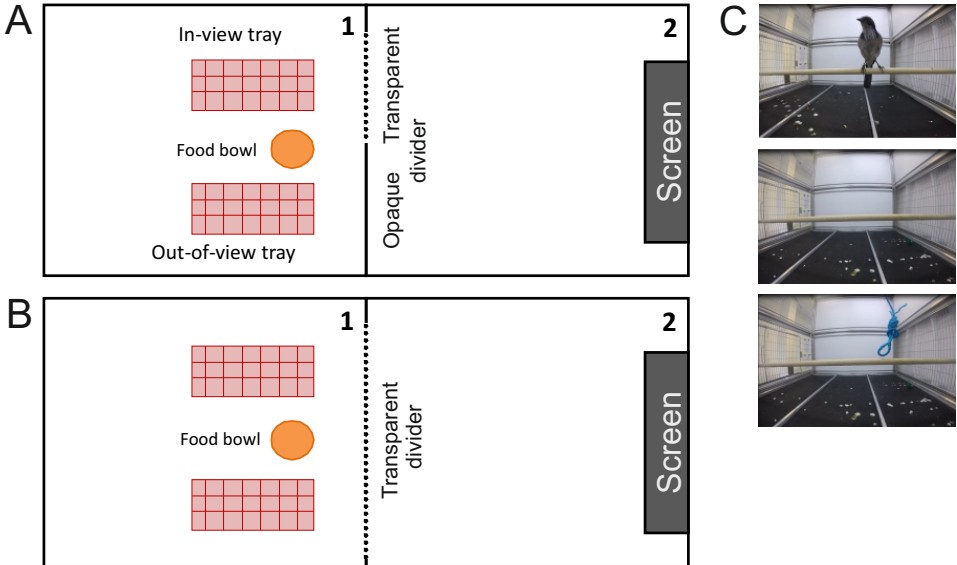

**Figure 1** **Aerial view of the testing setups and example still frames of the videos used.** (A) An aerial view of the testing setup for Experiment 1. The cacher had access to two caching trays (red rectangle) and a bowl of food (orange circle). One tray was "out-of-view" behind the opaque barrier (thick grey line) and the other one was "in view" behind a transparent barrier (dotted grey line). The computer screen was mounted to the far wall of cage 2. (B) An aerial view of the testing setup for Experiment 2. Here, both trays are "in-view" behind a transparent divider. (C) Still frames of the video clips, from the top: a conspecific facing towards the subject, an empty cage, and a rope hanging from the ceiling (non-social control).

in *Clayton, Dally & Emery, 2007*). During this 15-minute long retrieval phase, the screen was covered. Subsequently, the caching trays were removed, birds were released into their home cage and maintenance diet was returned.

In a pre-test, two trials were conducted to ensure that birds were comfortable with caching in the experimental set-up, particularly in the presence of the computer screen. In these pre-test trials, birds were given access to a bowl containing food items and a single caching tray, which was placed in one trial behind the opaque, and in another trial behind the transparent side of the Perspex barrier. Birds were allowed to cache for 15 min whilst the video on the computer screen showed an empty cage. The order in which the birds experienced the caching tray being behind the opaque or the transparent sides of the barrier was counterbalanced across birds. Trials were repeated until each bird cached in both trays, with a maximum of two trials per side.

### Material
The videos presented were 15-minute long, without sound, and consisted of a looped sequence of 4 to 5-second long recordings. Videos were filmed using a GoPro® Hero 4 Black with 60 frames per second and a resolution of 930p and presented with a Lenovo® Thinkpad Edge E330 (50 to 60 Hz refresh rate, graphic card Intel HD graphics 4000) on a 17″ portable Lenovo® LCD monitor (60 Hz refresh rate). Note that LCD pixels do not flash between frames, and hence, LCD monitors exhibit no refresh-induced flicker.

The screen with the video was mounted to the wall of cage 2 (see Fig. 1A). The videos presented either a conspecific facing the cacher (Observer condition), an empty cage (Empty Cage condition), or a rope hanging from the ceiling (Rope condition; see Fig. 1C). On screen, conspecifics were between 6.4 cm to 8 cm in height, corresponding to a visual angle of 3.3° to 4.2° (assuming the cacher stands on the caching tray).

Birds serving as actors in the Observer Condition were kept mildly hungry by removing all food from their cage 30 min prior to filming. Actor birds were presented with food behind the camera to ensure that they attended to the direction of the camera and thus would appear to be attending to the cacher during the test trials.

Actors were chosen based on their reaction to being filmed: we choose only birds that sat calmly on a perch in front of the camera. All actor birds were familiar to the focal birds and were of the same sex as the respective focal bird. Three sets of videos were shot with three different birds to prevent any effects being caused by study-irrelevant features of the video, such as the identity of the conspecific shown.

The caching trays were plastic ice-cube trays with $2 \times 7$ cubes moulds filled with corn kibble. These caching trays were made visually trial-distinctive with Lego Duplo® blocks attached to one long side of the tray. A bowl containing 50 food items (the type of food depended on the cacher's food preference) was placed equidistantly to both trays.

### Analysis

Items cached in the trays during the caching period were counted and recorded by the experimenter. To obtain one independent measure for the birds' preference for caching in one of the two locations ("in view" versus "out-of-view") in Experiment 1, the difference of the number of caches made in the "out-of-view" tray minus the number of caches made in the "in-view tray" ($D = \text{Cached}_{out-of-view} - \text{Cached}_{in\ view}$) was compared across conditions. If scrub-jays represented the conspecific observer in the video as a threat to their caches, this difference should be larger in the Observer condition than in the two non-social conditions, $D_{Observer} > D_{Empty\ Cage}$ and $D_{Observer} > D_{Rope}$.

Aligned rank transformed data (*Wobbrock et al., 2011*) were submitted to a repeated measures ANOVA with the within-subject factor Condition (Empty Cage vs. Observer vs. Rope). Alpha was set at .05. One-sided permutation tests were calculated using R package coin (*Hothorn et al., 2008*) as a planned contrast between the Observer condition and non-social conditions (Empty Cage and Rope, separately).

Additionally, we analysed the data using Bayesian statistics. A failure to reject the null-hypothesis using frequentist inferences cannot be used to draw conclusions about the absence of an effect (*Dienes, 2016*). Thus, we calculated a Bayes Factor (*BF*) using JASP Version 0.8.1.2 (JASP Team, 2016) with a Bayesian repeated measures ANOVA with the within-subject factor Condition (Empty Cage vs. Observer vs. Rope). Here, the null hypothesis was that Condition does not influence the preference for one of the two trays. A $BF_{10} > 3$ would suggest support for H1 (i.e., $D_{Observer} > D_{Empty\ Cage}$ and $D_{Observer} > D_{Rope}$) and a $BF_{01} < 0.333$ would suggest support for the H0 (i.e., $D_{Observer} = D_{Empty\ Cage}$ and $D_{Observer} = D_{Rope}$).

**Table 1  Median number of items cached in Experiment 1.**

| | Median items cached (Min-Max) | | |
| --- | --- | --- | --- |
| | **Total** | **Out-of-view** | **In-view** |
| *Condition* | | | |
| Observer | 6 (1–30) | 4 (0–8) | 2.5 (0–22) |
| Empty cage | 13 (1–39.5) | 5 (0–17) | 8 (0–22.5) |
| Rope | 9 (1.5–21.5) | 5 (0–16) | 6 (1–10) |
| *Trial No.* | | | |
| 1 | 5.5 (1.5–39.5) | 1 (0–17) | 4.5 (1–5–22.5) |
| 2 | 8 (1–24.5) | 4 (0–16) | 3.5 (0–17) |
| 3 | 9 (1–30) | 7 (0–9) | 9 (0–22) |

## Results

One male subject (Subject No. 210) was excluded from the analysis because he failed to cache for the duration of all test trials (final $n = 9$). Table 1 shows the median number of caches made in both trays. There was no main effect of Condition on the total number of caches made ($F(1, 16) = 1.34, p = .290$).

Critically, there was also no main effect of Condition on the difference of caches made in the "in-view" minus "out-of-view" tray ($F(2, 16) = 0.572, p = .575$; see Fig. 2). Planned contrasts showed that the difference of caches made in the "out-of-view" tray minus the "in-view" tray in the Observer condition (Median $= -1$, range $= 21$) was not higher than in the Empty Cage condition (Median $= -3$, range $= 9$; $Z = -0.043, p = .523, d = -0.014$), or the Rope condition (median $= 0.5$, range $= 19.5$; $Z = -1.003, p = .838, d = -0.337$).

In order to check whether the trial sequence had an effect on caching, we assessed the number of caches made in both trays across trials post-hoc. Table 1, lower part, shows the number of caches made per condition. There was no difference in number of caches made ($F(2, 16) = 0.021, p = .980$).

The Bayes factor was $BF_{10} = 0.279 \pm 0.572$, suggesting that our data support the null-hypothesis, i.e., a similar preference for the two trays across all three conditions. Thus, our data suggest that the scrub-jays did not adjust their cache-protection strategies in response to the video playback presented. When comparing the Empty Cage and the Observer condition with a Bayesian $t$-test, we found again that our data supported the null-hypothesis, $D_{\text{Observer}} = D_{\text{Empty Cage}}$, $BF_{10} = 0.331 \pm \sim 0.004$, whilst the comparison between the Rope and the Observer conditions showed only anecdotal evidence for the null-hypothesis, $D_{\text{Observer}} = D_{\text{Rope}}$, $BF_{10} = 0.780 \pm \sim 1.560e^{-4}$.

These findings suggest that the scrub-jays' caching behaviour was not affected by the presence of the conspecific displayed on the video screen: the birds were not more likely to employ cache protection strategies when presented with a video of **a** conspecific compared to the non-social controls.

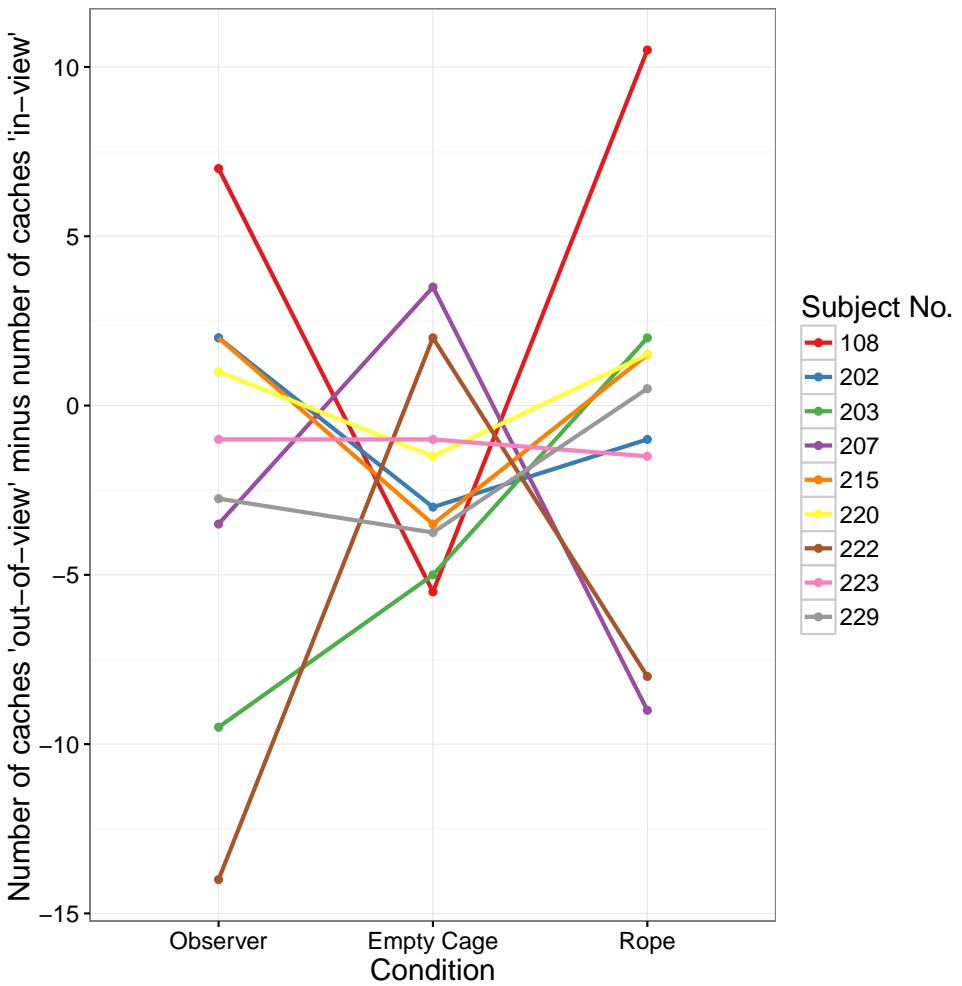

**Figure 2  Number of caches cached in the "out-of-view" tray minus the number of caches cached in the "in-view" tray in Experiment 1.** Each data point indicates the difference for one bird in each of the three conditions.

## EXPERIMENT 2

Experiment 2 addressed the question of whether the jays' re-caching behaviour was influenced by the different videos. Cachers had access to two caching trays, both "in-view", and, after a break of 15 min, had the opportunity to re-cache 'in private', i.e., in the absence of the observer, any caches made in the initial caching period. As in Experiment 1, birds were presented with different videos during caching, namely of an observer (social condition), as well as of an empty cage and a rope hanging from the ceiling (non-social conditions). The proportion of re-caches (number of re-caches divided by the total number of caches) made was investigated. If the scrub-jays were sensitive to the video presented on the video screen, they were expected to re-cache a larger proportion of initially made caches after having had cached in front of a video showing the observer (Observer condition) than

after having had cached in front of a video showing an empty cage (Empty Cage condition) or a rope hanging from the ceiling (Rope condition).

## Method
### Subjects

[1] One male bird, 210, was not tested because he failed to cache in Experiment 1.

Five female and four male[1] sexually mature scrub-jays were housed and tested under the same conditions as laid out in Experiment 1 between February and June 2015. All birds had previously participated in Experiment 1. The overall setup was the same as in Experiment 1, except that a fully transparent divider was placed between cages 1 and 2 such that there was no difference in the visual accessibility of the two caching locations, i.e., both caching trays were "in-view".

### Procedure and material

Each bird completed three test trials, separated into 3 phases, as described in Experiment 1. Importantly, during the 15-minute long re-caching phase, the video screen was covered, thus re-caching was in "private". Note that new videos were shot for this experiment to avoid habituation, and new actor birds were used, that is, the focal birds saw a different observer than in Experiment 1.

### Analysis

To ensure that birds could re-cache items, birds had to initially have cached at least one food item in each condition. In an equivalent manner to Experiment 1, we analysed the proportion of re-cached items (number of re-cached items divided by total number of items cached) with a repeated measures ANOVA (factor Condition) and one-sided permutation tests as planned contrasts.

Additionally, we calculated a Bayes Factor ($BF$) using JASP Version 0.8.1.2 (JASP Team, 2016) with a Bayesian repeated measures ANOVA with the within-subject factor Condition (Empty Cage vs. Observer vs. Rope). Here, the null hypothesis was that Condition does not influence the preference for one of the two trays. A $BF_{10} > 3$ would suggest support for H1 and a $BF_{01} < 0.333$ would suggest support for the H0.

### Results

Subject Nr. 215 was excluded from the analysis because she failed to cache across all test trials (final $n = 8$). Table 2 shows the median number of caches made and the proportion of re-cached items. There was no main effect of Condition on the number of caches made in the caching phase ($F(1, 14) = 1.133$, $p = .349$).

Crucially, there was no effect of Condition on re-caching, that is, birds did not alter the proportion of re-cached items depending on the different stimuli presented on the video, ($F(2, 14) = 0.29$, $p = .75$, see Fig. 3). Birds did not re-cache a larger proportion of caches in the Observer condition (Median = 0.44, Range 0.31–0.62) than in the Empty Cage condition (Median = 0.37, range: 0.22–1; $Z = -0.041$, $p = .520$, $d = 0.00$), or than in the Rope condition (Median = 0.42, range: 0.29–1; $Z = -0.371$, $p = .582$, $d = 0.105$).

Similarly to Experiment 1, we assessed whether the trial sequence had an effect on caching as well as on the proportion of re-caching. Table 2, lower part, shows the number

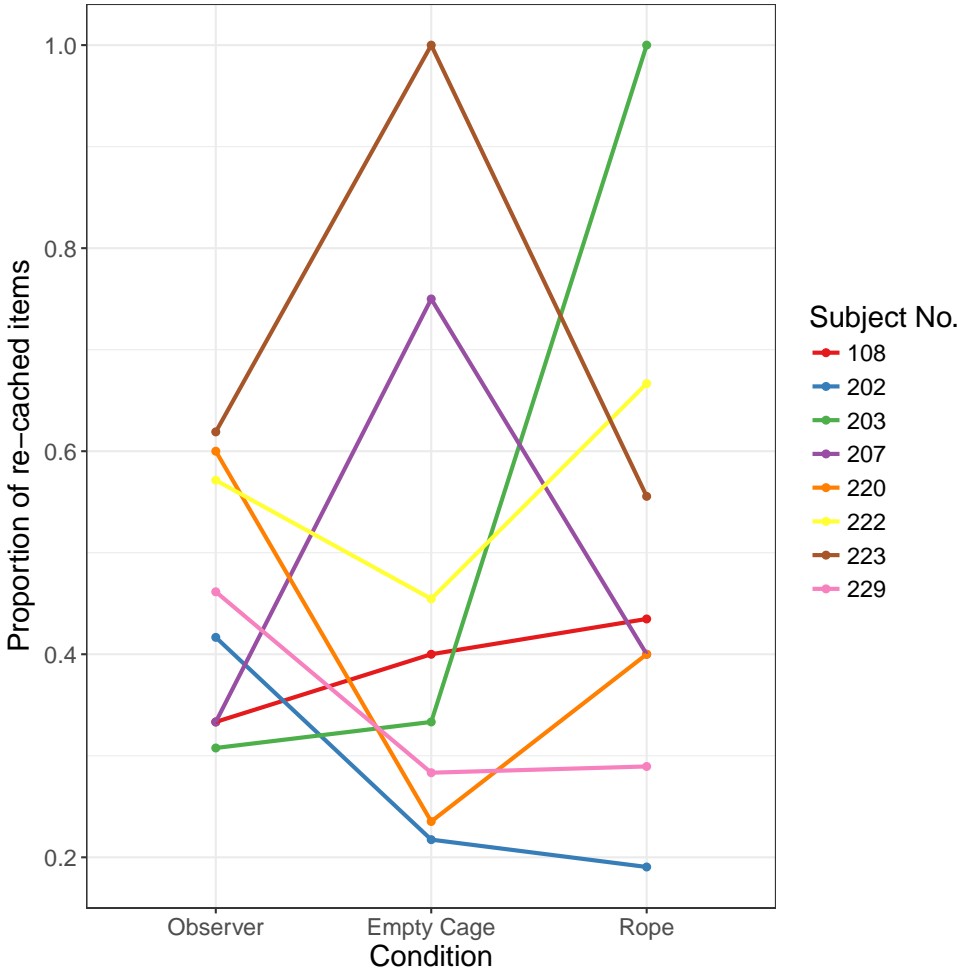

**Figure 3 Proportion of re-cached items in Experiment 2.** Each data point indicates the proportion for one bird in each of the three conditions.

**Table 2 Median number of items cached and re-cached in Experiment 2.**

|  | Median items cached (Min-Max) | Median proportion of re-cached items (Min-Max) |
|---|---|---|
| *Condition* | | |
| Observer | 12.25 (0–24) | 0.44 (0.31–0.62) |
| Empty cage | 6.25 (1–23) | 0.37 (0.22–1) |
| Rope | 6 (0–38) | 0.42 (0.29–1) |
| *Trial No.* | | |
| 1 | 12.25 (1.5–38) | 0.45 (0.19–1) |
| 2 | 11.75 (1-5–23) | 0.42 (0.22–1) |
| 3 | 8 (2.5–24.0) | 0.41 (0.28–0.6) |

of cached and re-cached items per trial number. There was no effect of trial number on the proportion of re-cached items ($F(2, 14) = 0.009$, $p = .991$). There was also no effect of trial number on the number of cached items ($F(2, 14) = 0.404$, $p = .675$).

The Bayes factor was $BF_{10} = 0.282 \pm 0.780$, suggesting that our data support the null-hypothesis, i.e., the amount of re-caches was similar across conditions. Comparisons using Bayesian t-tests showed that there was no difference in re-caches between the Observer and the Empty Cage conditions, $BF_{10} = 0.327 \pm \sim 0.014$ or the Observer and the Rope conditions, $BF_{10} = 0.268 \pm \sim 0.002$. Thus, our results suggest that the scrub-jays did not adjust their re-caching behaviour in response to the video playback presented.

## GENERAL DISCUSSION

In this study, we used California scrub-jays' cache protection strategies to assess the use of video playback when presenting social stimuli to corvids. Scrub-jays are known to protect their caches against potential pilferage by conspecific observers by caching out of the conspecific's view (*Clayton, Dally & Emery, 2007*; *Dally, Emery & Clayton, 2010*) or by re-caching previously made caches (*Emery & Clayton, 2001*). When presented with the video playback of a conspecific, scrub-jays did not employ protection strategies in either their caching or re-caching behaviours, suggesting that they do not respond to video playback in the same way they respond to live observers.

More specifically, in Experiment 1, the scrub-jays did not prefer to cache in an "out-of-view" tray over an "in-view" tray when presented with a video playback of a conspecific compared to the two non-social conditions. Similarly, in Experiment 2, the birds did not re-cache more when presented with a video of conspecific during an initial caching event compared to the non-social conditions. In a suite of previous studies using live observers with a very similar setup, the scrub-jays cached more in sites that were "out-of-view" of an observer and re-cached their caches once an observer has left the scene, while they did not show the same pattern of behaviours in a non-social condition, i.e., when they were caching in private (*Dally, Emery & Clayton, 2005*; *Dally, Emery & Clayton, 2010*; *Emery & Clayton, 2001*; *Thom & Clayton, 2013*). Although null results are difficult to interpret, our study suggests that the birds may not respond to video playback conditions in the same way as when confronted with a live conspecific in the context of caching. It should be noted that during the pre-test phase, the birds were shown an empty cage, albeit not the same one as in the test. However, familiarity with the empty cage video could explain why birds cached more in the empty cage condition (although this difference was not significant) in Experiment 1.

Consequently, contrary to the reports of successful use of video playback to display test stimuli to birds (e.g., *Galoch & Bischof, 2007*; *McQuoid & Galef, 1993*; *Ophir & Galef, 2003*; *Zoratto et al., 2014*), the birds in the present study did *not* adjust their behaviour to what was shown on the video. Especially relevant for this present study is the finding that rooks are able to differentiate between individual conspecifics presented on a video screen (*Bird & Emery, 2008*). In addition, we have previously shown that scrub-jays prefer to look at their conspecific partner compared to an object, regardless of whether the

conspecific was presented as video or live (*Brecht, 2017*). Hence, our results represent a noteworthy divergence from previous studies, considering that we provided the birds with videos similar to those successfully used in previous video playback experiments (e.g., *Bird & Emery, 2008*; *Guillette & Healy, 2017*). There are several cues that might be relevant in social interactions besides visual displays that were not present in the current study, such as olfactory and auditory cues, which, in real-life situations, can be used to recognise conspecifics (*D'Eath & Dawkins, 1996*). Compared to other bird species, however, corvids have a small olfactory bulb (*Bang, 1971*; *Healy & Guilford, 1990*) and are seen to use their sense of smell predominantly for detecting food (e.g., *Buitron & Nuechterlein, 1985*; *Harriman & Berger, 1986*). Additionally, there is so far no evidence that olfactory cues play a marked role in the social cognition of corvids. Hence, while it remains a possibility, given our current knowledge, it seems unlikely that the lack of olfactory cues might be responsible for the failure of the birds to adjust their behaviour to the video displays. In contrast to olfactory cues, auditory cues have been found to be of relevance when presenting video playback to birds (e.g., *Galoch & Bischof, 2007*). In other studies however, auditory cues were not necessary to prompt social learning (*Guillette & Healy, 2017*). In future studies, it will be important to manipulate both visual and acoustic cues when using video playback to investigate the role of each cue separately and of both cues together.

In our case, it seems likely that the absence of cache-protection strategies reported is associated with factors that might be especially important for the socio-cognitive nature of the caching context. One potentially important feature of a conspecific that was not available to the birds in this study specifically might be a real time response in the behaviour of the video conspecific. The relevance of a conspecific's behaviour has previously been reported in the successful recognition of and discrimination between conspecifics (*Keeling & Hurnik, 1993*; *McQuoid & Galef, 1993*; *Shimizu, 1998*). Similarly, it has been proposed that an absence of interaction with the focal bird deters hens from treating video stimuli as real conspecifics (*D'Eath & Dawkins, 1996*). Thus, the indifference towards the observer could have been due to the lack of behaviours from the video-observer: In Experiments 1 and 2, looped material was used in order to present the birds with a conspecific being in view and attentive throughout the whole video (as opposed to sleeping for example), which might have created an unusual deficiency of dynamic motion. In reality, it might be rare that a conspecific is facing towards a conspecific for a prolonged amount of time and does not interact with the cacher. Consequently, the altered video playback of conspecifics used here might not have been sufficient to elicit cache-protection strategies due to what they depicted. Hence, in order to guide complex decision making, such as a bird's caching behaviour, further research might need to include unaltered video playback.

While not the main focus of the present study, our results raise the additional question of *which* features of an observer trigger cache-protection strategies in corvids. One possibility might be the 'interaction' with the conspecific. In contrast to the dynamic relationship between a cacher and a live conspecific observer, the conspecific in the video did not respond to the cacher's behaviour in any way. This lack of interaction may have been an unusual situation for the birds being tested and might have led them to believe that their caches were safe, or at least that circumstances were 'unusual'. Relatedly, scrub-jays have

been shown to treat a mirror-image of themselves during caching as if they were in private, rather than as if the observed image in the mirror was a live observer. In other words, the 'conspecific' reflected by the mirror image was not sufficient to elicit cache protection strategies (*Dally, Emery & Clayton, 2010*). In addition, a cache-protection strategy that takes into account the current motivational state of an observer seems to be based on the behaviour of the observer at the time of caching (*Ostojić et al., 2017*). Consequently, it might be the contingent responses from an interactant that could serve as a salient cue that the birds use to employ cache protection strategies. Throughout their lives, the scrub-jays might have learned that conspecifics will respond to their caching behaviour, for example by coming closer to the caching site or by carefully monitoring the caching. Hence, further research is needed to assess which aspects of the *behaviour* of an observer are necessary conditions for cache-protection strategies to occur (*Ostojić et al., 2017*).

## CONCLUSION

In conclusion, our results suggest limitations in using video playback to present social stimuli to corvids. Depending on the context, different stimuli of a conspecific might be necessary in order to recognise conspecifics, and to behave accordingly. Specifically, there might be some limitations in regards to the use of video playback to mimic real-life situations involving social agents, such as being observed by a conspecific, to a sufficient extent to influence the birds' decision making such as their cache-protection behaviour.

## ACKNOWLEDGEMENTS

We thank Sam Melvin, Sarah Manley, and Ivan Vakrilov for animal husbandry. We thank two anonymous reviewers for helpful comments on a previous version of the manuscript.

### Funding

This work was supported by a PhD scholarship from the Cambridge Commonwealth, European and International Trust and Lucy Cavendish College Cambridge to Katharina Brecht. Ljerka Ostojić and Nicola Clayton received funding from the European Research Council under the European Union's Seventh Framework Programme (FP7/2007-2013) / ERC Grant Agreement No. 3399933, awarded to NSC. Edward Legg was supported by the Leverhulme Trust, Grant number RPG-2014-353. The funders had no role in study design, data collection and analysis, decision to publish, or preparation of the manuscript.

### Grant Disclosures

The following grant information was disclosed by the authors:
Cambridge Commonwealth, European and International Trust.
Lucy Cavendish College Cambridge.
European Research Council: 3399933.
Leverhulme Trust: RPG-2014-353.

## Competing Interests

The authors declare there are no competing interests.

## Author Contributions

- Katharina F. Brecht conceived and designed the experiments, performed the experiments, analyzed the data, prepared figures and/or tables, authored or reviewed drafts of the paper, approved the final draft.
- Ljerka Ostojić conceived and designed the experiments, analyzed the data, authored or reviewed drafts of the paper, approved the final draft.
- Edward W. Legg conceived and designed the experiments, authored or reviewed drafts of the paper, approved the final draft.
- Nicola S. Clayton conceived and designed the experiments, contributed reagents/materials/analysis tools, authored or reviewed drafts of the paper, approved the final draft.

## Animal Ethics

The following information was supplied relating to ethical approvals (i.e., approving body and any reference numbers):

The experiment was conducted under the UK Home Office project licence PPL 80/2519.

## Data Availability

Figshare: DOI 10.6084/m9.figshare.4055133.

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
