# Peer review of "Difficulties when using video playback to investigate social cognition in California scrub-jays (Aphelocoma californica)"

_PeerJ, doi:10.7717/peerj.4451_

## Round 0.1 · original submission · Major Revisions

· Academic Editor

Major Revisions

Two expert reviewers have provided some very helpful feedback on your MS. I agree with the reviewers that this MS presents some valuable information regarding the potential use of video playback in these sorts of experiments, but I also agree with their concern that the limitations of the current study may limit the generalizability of your conclusions. I agree with both reviewers that it is very important to specify the conditions of the video; for example, were there auditory cues signaling the presence of an observer? Could the birds on the video be perceived as looking directly at the caching birds? How important are olfactory cues in leading the birds to respond as if an observer was present? Was the conspecific in the video familiar to the caching bird? I also agree that the experiment would be stronger if you could directly compare to a live observer condition with the same birds. Please prepare a revision of the MS that takes into account the reviewer’s useful suggestions and prepare a response letter indicating how you have done so within the revised MS. Please be sure to resolve the discrepancy in reporting the sample size as noted by Reviewer 1.

I have a few additional small points:

In looking at Figure 2, it does appear as if the birds treated the rope condition more like the observer condition, compared to the empty cage condition. Movement then is maybe an important cue – can you comment more on this, even though the results are not significant? I am a bit concerned with the small sample size and the very different patterns shown by the birds that it may not be the most informative to collapse the results together but rather to look at individual patterns. The same issue is apparent in Figure 3 for Exp. 2.

Were you able to examine any instances of re-caching in Experiment 1?

Delete “or not” in the third line of the abstract and also line 17, 67 and so on.

Something is wrong with line 30. Please re-phrase.

On lines 66-67, move “only” to after “involved”

Please insert commas after “e.g.” and “i.e.”

Reviewer 1 ·

Basic reporting

This study investigates whether scrub-jays change their caching behaviour in response to video playback of a conspecific. In contrast to previous studies with live observers, video playback of an observer did not influence scrub-jays’ cache protection strategies. This suggests that video playback might not be an appropriate method to simulate conspecifics in this context.

The paper is generally very well written and easy to read. Research questions and predictions have been stated clearly in the introduction, and relevant literature is cited. Results have been reported clearly and the discussion is logical. Figures are relevant and easy to interpret, although the pictures of the video clips (Figure 1c) could be bigger. The raw data has been supplied and it is sufficiently described.

Experimental design

Research question is well defined and relevant: video playback could be a useful tool to manipulate social stimuli in behavioural experiments, but we don’t know if corvids perceive conspecifics in the video same way that they perceive live conspecifics.

For most part, the methods have been described with enough detail (see detailed comments below for few clarifications).

Experiment has appropriate controls. Including fourth treatment with live conspecific would have made the results stronger, as it would have been possible to compare birds’ response to live observer to their response to video playback. However, I don’t think that this is a big issue, as many previous studies (with similar methods) have shown that birds change their caching strategies when observed by live conspecifics.

Another weakness of the study is a small sample size. I understand the difficulty of getting a big sample size in these kinds of experiments, but as there seem to be a lot of variation among individuals (Figures 2 and 3), it is difficult to make conclusions based on the number of individuals tested.

Detailed comments

The sample size in the first experiment is unclear. First it is stated that 10 individuals were tested (line 117). Then it is said that two birds died (lines 122-124), which resulted the sample of 6 scrub jays (shouldn't this be 8?). However, in the results (lines 200-201) the final sample size is stated to be 9. Could you please clarify this?

In both experiments individuals completed three test trials. How much time there was between the trials?

Lines 125-135: Did the videos have sound?
Lines 139-140: Three different birds were filmed for video playback. Were these individuals familiar to the birds that were tested? Were they females or males?
Lines 254-255: Did you use the same birds as in the Experiment 1, or did individuals see a different observer in the second experiment?

Validity of the findings

The study did not find evidence that scrub-jays changed their caching behaviour in response to videos of a conspecific. As authors mention in the discussion, these null results are difficult to interpret, and could be the result of many factors, such as the looped material used in the videos. In addition, small sample size and high individual variation make it difficult to interpret the results.

I don’t think that the study provides clear evidence that video playback could not be a useful tool in these kinds of experiments, as its efficacy could depend on the content and presentation of the video. However, authors have discussed these issues and have not overstated their findings. Overall, the study provides relevant information on the challenges to use video playback in this context.

Additional comments

Were there any differences in the total number of caches (in-view + out-of-view caches) among video treatments? I would suggest including this test in the results. Also, did you test if trial order had an effect on the results?

In the discussion, authors mention that one explanation for the results could be the lack of interaction between an observer and a cacher, and I agree that this could be important. Did you see any of the cachers trying to interact with observers (vocalize, for example)? Also, did you look at any other behaviours during video playback, besides caching? For example, did birds pay more attention (e.g. spent more time looking at the screen) to video playback of an observer, compared to the controls?

When reporting the results of the experiment 1, please provide both minimum and maximum number of caches for each treatment (now range seems to include only maximum values?). In the results of experiment 2, medians and range for each treatment have been reported twice (lines 280-282 and 286-288), which is unnecessary.

Minor things
- I think that the statistics should be in brackets (e.g. lines 206-207, 210, 211 and so on).
- The subheading in the first experiment is “Results and Discussion” and in the second experiment only “Results”. Please change one of them, so that they are consistent.
- Line 30: change allows to allow
- Line 41: change investigating to investigation
- Line 57: I would suggest citing here also a more recent video playback experiment: Guillette, L. M., & Healy, S. D. (2017). The roles of vocal and visual interactions in social learning zebra finches: A video playback experiment. Behavioural processes, 139, 43-49.
- Line 280: change re-cache to re-cached
- Figure 2. Change y-axis title “Number of cached” to “Number of caches”
- Some of the references are missing DOI.

Reviewer 2 ·

Basic reporting

The current manuscript examines the use of video playback in California scrub jay caching behaviour. The main objective is to test is video playback produces the same results as in previous work. To do this, the authors set out to replicate previous findings—that: 1) California scrub jays, in the presence of a conspecific, preferentially cache in ‘out-of-view’ locations; and, 2) demonstrate a higher propensity to re-cache—under a video playback context in two experiments. In both Experiment 1 and 2 birds were tested under three treatments: 1) observer: video playback of a cage with a conspecific; 2) empty cage: video playback of a cage with no conspecific; and, 3) rope: video of a cage with a rope attached to the ceiling. Experiment 1 examined the relative difference in the number of caches made by birds behind an opaque and a transparent barrier across treatments. Experiment 2 examined the proportion of re-caches made by birds across treatments. Analyses showed that neither cache ratio transparent:opaque barrier did not change significantly across condition (Exp 1) nor re-cache propensity (Exp 2).

General comments:

There needs to be more information included to make this paper interesting to a wider audience. Can you include information about what other corvid cachers do? There has been work on other caching species in relation to social conditions e.g. Clary & Kelly 2011 Anim. Cog. There has also been some recent work on comparing results from live and video demonstrators in a social learning task in the zebra finch e.g. Guillette et al. 2017 Behav. Process. and in using video playback in foraging in great tits Hamalainen et al. 2017 PeerJ. There is an overall light touch with the introduction and many relevant papers are not included. Please expand the intro and discussion.

I am not sure if this paper should be published with the conclusion that 'videos may not be an appropriate tool to manipulate social stimuli in the context of cache protection strategies (abstract)' because of the null results with a low N and, moreover, because unaltered video playback was not used.

Experimental design

Unclear if the rope in test condition 3 was indeed moving? Please clarify.

L129-130 Is this height within the normal range for jays at this distance? Please clarify.

L135 was the sound in the video on? who was the demonstrators? Male? Female? Familiar? Dominant? Please add this information.

L173 how many trials per day?

L303 I would not rule out this method, as you should try unaltered playback and a larger sample size.

Validity of the findings

If the same birds were used in both experiments wouldn’t they have learned from the first experiments that their caches are never pilfered which could lead to low to no re-caching in the second experiment? Hasn’t previous work found that birds need experience being pilfered to learn to re-cache?

Unclear why in Experiment 1 only the difference in cache number between the two sites of interest was analysed – why not also look at the difference in the number of caches across test conditions? Condition might, for example, affect caching propensity, even if it does not affect ‘preference’ in cache location. Can the authors please include this information?

It is unclear what the sample size was (see specific comments for line 124, below) but if it was 6, then that may be too few, especially for null results. Can the authors please not only clarify how many subjects were run per experiment (and how many were used in the analyses), but also include a power analysis from the published work of the effect size for the study that they are currently replicating? How many individual were included in the original study?

Additional comments

The methods section requires restructuring to present details in a more concise/non-repetitive manner. Please include a general Subject, Housing and Husbandry section, followed by an apparatus section to detail the testing set-up. Then introduce each experiment, and when doing so, please limit procedural details to the experimental procedure section i.e., remove large paragraphs under Experiment 1 and 2 headers.

L30 – some words missing e.g. ‘for a’ contrast ‘between’

L31 insert ‘using video clips’

L42 Please remove ‘sophisticated’ here and elsewhere, as this (and other similar/frequently used terms e.g., ‘complex’) was not measured.
L45 Please consider adding a full stop after ‘detail’.

L5-60 this info is from a PhD thesis, I’m not sure about the journals policy of citing as such. Maybe ‘Pers Comm’ is more appropriate? Same for line 325.

L63-4 I’m not following the reasoning here that the bird may not interpret the video as a social stimulus. Please clarify/expand.

L76-80 Please consider moving these examples into the previous paragraph, making L81-89 the final paragraph describing the current study’s aims.

L82 – who is ‘they’ referring to? The observer? The focal animal? Please clarify.

L91-115 More or less states the same information detailed in the Procedure section L149-175 below. Please streamline such that this information is presented in a concise and non-repetitive form.

L95-97 These data are already detailed in the introduction, please remove.

L107-109 Same comment as above.

L109-111 These predictions are stated previously in L99-102 and are therefore repetitive, please remove.

L124 Sample size of six? But 10 birds – 2 lost = 8. And similarly, in L201 and Figure 1 n = 9. Please clarify.

L202-205 Please consider putting this information in a Table. Same suggestion for 280-282

L196-197 Unclear how the Bayesian analysis (BF) would ‘suggest’ support i.e., one either accepts or rejects the null. Please clarify and/or remove this term.

L215 ‘some evidence’ – but this number is <0.333, as is the BF value for Empty Cage versus Observer i.e., both tests support the null hypothesis, but when worded as such the reader is lead to believe there is other evidence to the contrary (when there isn’t). Please reword.

L228-241 Please see comment for L91-115

L234 are these the same birds as in the previous experiment. If so, hadn’t they learned that their caches will not have been pilfered from Exp 1.?

L 286 how many seeds were eaten throughout all experiments? Would this be useful information?

---

## Round 0.2 · Minor Revisions

· Academic Editor

Minor Revisions

Decision on Brecht et al.

Thank you for such a detailed response to the reviewers’ comments during the last round of reviews. I was able to obtain a review on the revised MS from one of the previous reviewers, who was, for the most part, satisfied with the revision. I agree that the revisions have improved the readability and clarity of the MS. However, this reviewer does raise an interesting point about the familiarity of the empty condition relative to the other conditions in Exp. 2. It might be worth including a brief comment about that. I realize that the video itself was new, but perhaps the context being familiar made it less novel compared to the other two conditions, as the reviewer suggests.

There are some other very minor revisions that I’d like you to attend to before I can formally accept the MS.

Please do check the reference list carefully to make sure all cited works are included.

It isn’t quite clear to me how it was decided which birds would be actors and which would be used in videos and why three different sets of videos were used. Please provide a bit more detail here.

It is not made explicit in the methods that each bird was exposed to each condition X number of times (once?) and that the order of presentation of trial types was randomized (if so) or counterbalanced. This can maybe be assumed from reference to the pseudorandomization of trial order on lines 175-176 but you never state clearly that each trial is a different type of trial rather than a repetition of the same stimulus.

Please be very clear about the within subjects design in the procedure section.

Some other very minor edits:

Please place a comma after stimuli on line 38.
On line 40, investigating should be investigation.
Line 55, instead of “to display conspecifics in birds”, it might sound better to say “conspecifics to birds…”
Guillette is still misspelled on line 60. Please check carefully throughout.
The paragraph beginning at the bottom of the first page of the introduction and continuing onto the third page is very long. Consider breaking this up into separate paragraphs (with breaks perhaps at line 66 and line 74).
Line 239, please place “a” before “conspecific” or pluralize.
On line 379, change “life” to “lives”.

Thank you for submitting such interesting and potentially important work to PeerJ.

Reviewer 1 ·

Basic reporting

The manuscript is now much improved. Especially introduction and discussion are more comprehensive and authors have described the methods in more detail.

Experimental design

The experimental procedure and the conditions of video playback are now much clearer.

I have only one comment regarding the pre-test trials before the experiment (lines 166-174). Did I understand right that during these trials birds saw the same video of an empty cage that they saw later during the actual experiment (empty cage condition)? If so, birds were already familiar with this video, whereas two other conditions (rope and observer) were novel. Do you think that this could have influenced birds’ behaviour? It seems that in Experiment 1, the total number of caches was highest in empty cage condition (although this is not significant), and I am wondering if this could be explained by the fact that birds were already familiar with that stimulus?

Validity of the findings

No comment

Additional comments

The authors have responded well to my and other reviewer’s comments and done a good job revising the manuscript. I have only few minor comments:

Please check that all the papers that you are citing are included in the reference list. At least Silwa and Freiwald, 2017; Bugnyar, 2011; Bugnyar & Heinrich, 2005; and Guillette & Healy, 2017 are missing from the references. Also, Guillette is misspelled (line 60, 333 and 345).

Line 278: I believe that you are referring here to Table 2.

Line 342: misspelling birds

---

## Round 0.3 · accepted · Accept

· Academic Editor

Accept

Thank you for attending to the last very minor set of issues. I am happy to now accept your paper for publication in PeerJ. However, at proofing, please do be sure to change "choose" to "chose" on line 190.